# Adapting to Reward Progressivity via Spectral Reinforcement Learning

**Michael Dann & John Thangarajah**
School of Computing Technologies, RMIT University, Melbourne, Australia
`{michael.dann,john.thangarajah}@rmit.edu.au`

## Abstract

In this paper we consider reinforcement learning tasks with *progressive rewards*; that is, tasks where the rewards tend to increase in magnitude over time. We hypothesise that this property may be problematic for value-based deep reinforcement learning agents, particularly if the agent must first succeed in relatively unrewarding regions of the task in order to reach more rewarding regions. To address this issue, we propose *Spectral DQN*, which decomposes the reward into frequencies such that the high frequencies only activate when large rewards are found. This allows the training loss to be balanced so that it gives more even weighting across small and large reward regions. In two domains with extreme reward progressivity, where standard value-based methods struggle significantly, Spectral DQN is able to make much farther progress. Moreover, when evaluated on a set of six standard *Atari* games that do not overtly favour the approach, Spectral DQN remains more than competitive: While it underperforms one of the benchmarks in a single game, it comfortably surpasses the benchmarks in three games. These results demonstrate that the approach is not overfit to its target problem, and suggest that Spectral DQN may have advantages beyond addressing reward progressivity.

## 1 Introduction

In decision making tasks involving compounding returns, such as stock market investing, it is common for the rewards received by the agent to increase in magnitude over time. An investor that starts out with little capital will achieve relatively small profits or losses early on, but if they are successful, their capital – and hence their potential profits and losses – will gradually increase. This property also arises in many games settings. For example, in the *Atari* game *Video Pinball*, the player can increase a "bonus multiplier" that increases the score paid per bumper hit. Since the bonus multiplier does not reset until the player dies, the rewards typically increase with time. In this paper, we refer to any task exhibiting this phenomenon as a *progressive reward task*.

We hypothesise that reward progressivity may be problematic for value-based deep reinforcement learning agents. Our rationale is that the temporal difference errors that arise under such algorithms typically scale with the magnitude of the training targets. Accordingly, experience collected from states with large expected returns may dominate training, causing the agent's performance to degrade in other states. While this is rational in one sense – generally, it is more important to make accurate decisions in high-stakes situations – it is potentially harmful on progressive reward tasks. This is because, on progressive reward tasks, *it may be necessary for the agent to perform well in relatively unrewarding regions before it can reach more rewarding regions*. For example, in stock market investing, an investor can only reach states where they have large capital if they first perform well with small capital. Note too that the rewards need not be strictly progressive for this problem to arise; all that is needed is for the rewards to be progressive over some extended period.

Algorithms that clip the reward, such as *DQN* (Mnih et al., 2015), are less susceptible to this problem, because they do not perceive increases in reward magnitude beyond the clipping point. However, it is straightforward to construct examples where reward clipping masks the optimal solution. For example, in *Bowling*, clipping the rewards to [-1, 1] makes bowling a strike appear no better than hitting a single pin (Pohlen et al., 2018). While subsequent approaches make learning feasible without reward clipping (van Hasselt et al., 2016; Pohlen et al., 2018), we show in Section 3 that

these methods are negatively impacted by strong reward progressivity. Motivated by this, we seek an algorithm that is capable of learning from unclipped rewards, while mitigating the negative impact that large rewards have on the agent's ability to learn in less rewarding regions of the task.

To meet this aim, we propose *Spectral DQN*. Under this approach, rewards are decomposed into a multidimensional *spectral reward*, where each component is bound to [-1, 1]. The upper frequencies of the spectrum activate only when large magnitude rewards are received. For example, in *Video Pinball*, the lowest frequency activates on any score, while the upper frequencies only activate when the player has accumulated a large bonus multiplier. The bounded nature of the spectral rewards allows the agent to learn expected *spectral returns* without instability arising from large training targets. Meanwhile, the full, unclipped expected return can be recovered by summing across the return spectrum. (The full, technical definitions of these terms are provided in Section 4). In terms of addressing reward progressivity, a key advantage of this approach is that it allows us to balance the training loss and prevent the upper frequencies from receiving undue weight.

To test whether this approach helps mitigate the impact of reward progressivity, we perform two set of experiments: First, we apply Spectral DQN to two domains that we specifically designed to exhibit strong reward progressivity. While previous methods struggle on these tasks, failing to learn almost anything at all in the more extreme domain, Spectral DQN performs markedly better, making considerable progress on both tasks. Next, we apply our approach to a less constructed set of tasks; namely, 6 standard *Atari* games. Only some of these games exhibit reward progressivity, and none of them nearly so strongly as the extreme domains from earlier. While Spectral DQN does not outperform the benchmarks as noticeably in these domains (it is beaten by one of the benchmarks in a single game, but comfortably outperforms both benchmarks in three games), the results clearly show that Spectral DQN is not overfit to constructed domains. Moreover, since its outperformance in some of the games cannot be attributed solely to reward progressivity, it appears that the approach may offer additional advantages, as explained later in our analysis of the results.

## 2 PREVIOUS APPROACHES TO HANDLING REWARD VARIABILITY

In the *Atari* domain, where the DQN algorithm was first evaluated, the agent receives a reward equal to the score increase at each frame. However, since score magnitudes vary greatly across games, DQN clips all rewards to $[-1, +1]$. This constrains the size of the training targets so that it is easier to find a step size that performs well across the entire suite of games. While this heuristic turns out to perform well in *Atari*, a clear downside of the approach is that the agent becomes unable to distinguish large rewards from small rewards.

The *Pop-Art* algorithm (van Hasselt et al., 2016) offers a more principled solution. It trains from the unclipped reward, while normalising the training targets to have zero mean and unit variance. To ensure that the network's predictions are preserved when the normalisation parameters are updated, the output of the final layer is scaled and shifted by an offsetting amount.

Pohlen et al. (2018) propose an alternative approach to handling unclipped rewards that reduces the variance of the training targets by applying a squashing function, $h$. The agent learns squashed action-values, $\tilde{Q}(s, a) = h(Q(s, a))$, via the transformed Bellman backup:

$$\tilde{Q}(s_t, a_t) \leftarrow \tilde{Q}(s_t, a_t) + \alpha \big[ h\big(r + \gamma \max_{a'} h^{-1}(\tilde{Q}(s_{t+1}, a'))\big) - \tilde{Q}(s_t, a_t) \big] \tag{1}$$

The authors prove that the transformed Bellman operator remains a contraction provided that both $h$ and $h^{-1}$ are Lipschitz continuous and $\gamma < 1/L_h L_{h^{-1}}$ (where $L_h$ and $L_{h^{-1}}$ are the respective Lipschitz constants). In their experiments, they use the following squashing function:

$$h(x) = sign(x)(\sqrt{|x| + 1} - 1) + \epsilon x \tag{2}$$

where $\epsilon > 0$ ensures that $h^{-1}$ is Lipschitz continuous.

For the remainder of this paper, we refer to this approach as *target compression*. In the *Atari* domain, Pohlen et al. found target compression to perform significantly better than Pop-Art. It has since become the dominant method for handling unclipped rewards in *Atari*, with several recent large-scale agents (Kapturowski et al., 2019; Badia et al., 2020a;b), including the state-of-the-art *Agent57*, favouring target compression over Pop-Art.

## 3  DEMONSTRATING THE PROBLEM OF REWARD PROGRESSIVITY

While both Pop-Art and target compression address reward variance *across* tasks, making it easier to devise a single learning configuration that performs well across a task suite, they were not designed to address *intra-task* reward variability. As such, we posit that they are ill-suited to addressing reward progressivity, since it is an intra-task property. To test this claim, we introduce two domains that are designed to exhibit extreme reward progressivity, and thus exacerbate the issue: *Ponglantis* and *Exponential Pong*.

**Ponglantis.** In *Ponglantis*, the game starts out exactly the same as the *Atari* game *Pong*, where valued-based deep reinforcement learning methods are known to perform well (Mnih et al., 2015). The reward scale in *Pong* is small relative to most *Atari* games; the player receives a +1 reward for hitting the ball past the computer paddle, and a -1 reward if they concede a point. In *Ponglantis*, however, if the player manages to score 10 points, the task immediately switches to *Atlantis*, where the reward scale is roughly 1000 times larger. The *Atlantis* phase ends upon the loss of a single life.

**Exponential Pong.** The game *Exponential Pong* is identical to regular *Pong*, except that the rewards are multiplied by $2^n$, where $n$ is the player's current score.

We applied two agents to these games: Pop-Art and DQN+TC, where the latter is a version of DQN that we modified to use target compression and learn from the unclipped reward. The results of this experiment, averaged over 5 seeds, are shown in Figure 1. (For full experimental details, see the Appendix.) To simplify analysis of the *Exponential Pong* results, we have plotted the unexponentiated scores, i.e. the scores that the agents would have achieved under standard scoring in *Pong*. For context, the best possible score on standard *Pong* is +21, and the worst possible score is -21.

In *Exponential Pong*, which exhibits the more extreme reward progressivity (since the reward multiplier can potentially reach $2^{20} \approx 1$ million), both agents essentially fail, providing strong support for our hypothesis that reward progressivity can be damaging, at least in extreme cases.

In the more moderate case of *Ponglantis*, Pop-Art fails to make meaningful progress beyond the low-scoring *Pong* phase of the task. The likely reason for this is that as soon as the agent starts reaching *Atlantis* and receiving large rewards, the target scaling mechanism employed by Pop-Art starts dividing the training targets by a large factor, heavily dampening the much smaller learning signals arising from the *Pong* phase of the task. On the other hand, DQN+TC is eventually able to make some progress into the *Atlantis* phase. The fact that DQN+TC outperforms Pop-Art is consistent with our hypothesis; while Pop-Art sees the reward scale jump by a factor of $\approx$1000 upon reaching *Atlantis*, the training targets of DQN+TC jump by a smaller factor, owing to the square root squashing function (Equation 2). In Appendix B.2, we perform some additional experiments in *Ponglantis* to find where the "cutoff points" for Pop-Art and DQN+TC lie, i.e. how big the jump in reward magnitude has to be for the agents' performances to start deteriorating.

Given the above discussion, a natural question to ponder is whether DQN+TC might achieve better performance in *Exponential Pong* by employing a logarithmic squashing function instead of a square root. Unfortunately though, as Pohlen et al. (2018) point out to a reviewer of their work (see `https://openreview.net/forum?id=BkfPnoActQ`), a logarithmic transform does not have a Lipschitz continuous inverse, and thus the transformed Bellman is no longer guaranteed to be a contraction.

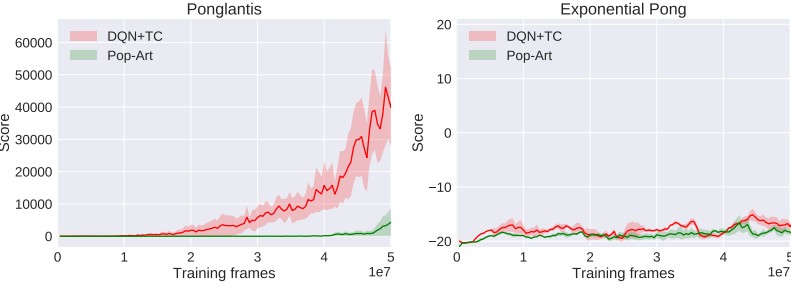

Figure 1: Mean episodic return for agents trained on *Ponglantis* and *Exponential Pong*.

## 4   SPECTRAL REINFORCEMENT LEARNING

Based on the findings of the previous section, it seems that what we require is an algorithm where:
(1) The training targets are constrained in magnitude, no matter how large the reward scale grows.
(2) Regions of the task with small rewards yield meaningfully large temporal difference errors, even
once much larger rewards have been discovered. Note that Pop-Art fails to meet the second criterion,
while target compression fails to meet both, albeit it takes larger shifts in reward magnitude for the
problems associated with reward progressivity to become apparent.

In this section, we introduce a reinforcement learning approach that meets both of the above criteria
by leveraging the idea of spectral decomposition. This algorithm does not in and of itself address re-
ward progressivity; in fact, we show that it behaves identically to the standard *Q-learning* algorithm
(Watkins, 1989), given certain caveats. However, the properties of this algorithm *facilitate* adaption
to progressive rewards, and in Section 5 we present a deep learning version of the approach that is
capable of learning in domains with extreme reward progressivity, such as *Exponential Pong*.

### 4.1   SPECTRAL DECOMPOSITION

*Spectral decomposition* is a broad term that arises in many different fields, but in this paper it refers to
a specific method for decomposing real numbers into a sum of exponentially weighted components.

The intuition behind our method is illustrated in Figure 2. In this example, the number being spec-
trally decomposed is 6.5. The method begins by dividing the number line into buckets of width $b^i$,
where $b > 0$ is the base of the decomposition and $i \in \mathbb{Z}_{\geq 0}$. Since the base here is 2, bucket 0 has
width 1 and spans from 0 to 1, bucket 1 has width 2 and spans from 1 to 3, bucket 2 has width 4 and
spans from 3 to 7, etc. The idea is that each bucket corresponds to a different frequency in the spec-
trum. The spectral decomposition is found by calculating the proportion of each bucket that is filled.
Since the number 6.5 completely fills the first two buckets, then 87.5% of the next bucket, then none
of the remaining buckets, the spectral decomposition of 6.5 to base 2 is $(1, 1, 0.875, 0, 0, \dots)$.

More generally, let $r$ be a non-negative real number, and let $r_b^{i+}$ denote the $i$th element of $r$'s spectral
decomposition to base $b$. The formula for calculating each element is:

$$r_b^{i+} = clamp\big((r - \tfrac{b^i-1}{b-1})/b^i, 0, 1\big) \tag{3}$$

(See the Appendix for the derivation of this formula.) The value $r$ can be recovered from the spectral
decomposition by multiplying the bucket widths by the bucket fill proportions and summing:

$$r = \sum_{i=0}^{\infty} b^i r_b^{i+} \tag{4}$$

This method can be extended to negative numbers by negating the sign on each element of the
representation. Given any real number $r$, the formula for calculating each element becomes:

$$r_b^i = sign(r) \times clamp\big((|r| - \tfrac{b^i-1}{b-1})/b^i, 0, 1\big) \tag{5}$$

### 4.2   DECOMPOSING THE RETURN

The core idea behind our reinforcement learning approach is to apply spectral decomposition to the
reward stream, and thus learn a spectral decomposition of the expected return. Before delving into
the technical details, see Table 1 for a worked example that highlights the intuition. The discount

| | 0 | 1 | 2 | 3 | 4 | 5 | 6 | 7 | 8 | 9 | 10 | 11 | 12 | 13 | 14 | 15 |
|---|---|---|---|---|---|---|---|---|---|---|---|---|---|---|---|---|
| Bucket number, i | 0 | 1 | | | 2 | | | | | | | 3 | | | | |
| Bucket width (=2$^i$) | 1 | 2 | | | 4 | | | | | | | 8 | | | | |
| Filled amount | 1 | 2 | | | 3.5 | | | | | | | 5 | | | | |
| Filled proportion | 1 | 1 | | | 0.875 | | | | | | | 0 | | | | |

⇒ Spectral representation of 6.5 = (1, 1, 0.875, 0, 0, 0, ...)

Figure 2: A worked example illustrating the spectral decomposition of the number 6.5.

| Time step, $t$ | $\gamma^t$ | Reward, $r$ | Spectral reward components | | | |
|---|---|---|---|---|---|---|
| | | | $r_2^0$ | $r_2^1$ | $r_2^2$ | $r_2^3$ |
| 0 | 1 | 1 | 1 | 0 | 0 | 0 |
| 1 | 0.99 | 4 | 1 | 1 | 0.25 | 0 |
| 2 | 0.98 | 11 | 1 | 1 | 1 | 0.5 |
| 3 | 0.97 | -4 | -1 | -1 | -0.25 | 0 |
| 4 | 0.961 | -10 | -1 | -1 | -1 | -0.375 |
| Return | | **2.254** | | | | |
| Spectral returns | | | 1.039 | 0.039 | 0.024 | 0.130 |
| Weight, $b^i$ | | | 1 | 2 | 4 | 8 |
| Spectral return × weight | | | 1.039 | 0.078 | 0.098 | 1.039 |
| $\Sigma$(Spectral return × weight) | | | **2.254** | | | |

Table 1: An example of a return being decomposed into a weighted sum of spectral returns.

factor is $\gamma = 0.99$, and the return over the first five time steps is 2.254. The usual way of calculating the return is, of course, to multiply the rewards by $\gamma^t$ then sum. The alternative, spectral approach is to first compute the spectral decomposition of the rewards according to Equation 5, then calculate the returns arising from each reward frequency. We refer to the decomposed rewards as *spectral rewards*, and to the decomposed returns as *spectral returns*. The overall return of 2.254 can be found by multiplying the spectral returns by their corresponding weights of $b^i$ then summing.

To appreciate why we go to this extra effort in calculating the return, note that the spectral rewards are all bound to [-1, 1]. This in turn constrains the magnitude of the spectral returns to be no more than $1 / (1 - \gamma)$, which, again, is useful from a function approximation standpoint. Moreover, note that the relative sparsity of large rewards is made more apparent by this approach: As we progress from $r_2^0$ to $r_2^3$, the magnitudes of the spectral reward components decrease, with only the two largest magnitude rewards triggering a response in $r_2^3$.

### 4.3 Spectral Q-learning

To translate the above intuition into a concrete algorithm, we begin by defining the *spectral action-value function* as:

$$Q^\pi(s, a, i) = \mathbb{E}_\pi[\sum_{k=t}^\infty \gamma^{k-t} r_{b,t}^i \mid a_t = a, s_t = s] \tag{6}$$

where $r_{b,t}^i$ denotes the $i$th element of the spectral reward at time $t$. As shown in the Appendix, it is straightforward to prove via the linearity of expectation that the standard action-value function can be recovered as the weighted sum:

$$Q^\pi(s, a) = \sum_{i=0}^\infty b^i Q^\pi(s, a, i) \tag{7}$$

This leads to a natural extension of the Q-learning algorithm (Watkins, 1989), whereby the greedy action is found by maximising this expression with respect to the available actions, and $Q^\pi(s, a, i)$ is trained via the standard Bellman backup, except that the reward is replaced by the $i$th element of the spectral reward. For clarity, the full pseudocode for this algorithm, which we term *Spectral Q-learning*, is provided in the Appendix (see Algorithm 1).

An important practical limitation of this algorithm is that it is impossible to train over full, infinite spectrum; that is, one must choose a finite upper bound for $i$. (This would not be an issue if we knew the maximum reward magnitude in advance, but for generality we do not wish to assume this.) Regardless, for base $b$, observe that the first $N + 1$ frequencies capture all rewards up to magnitude $(b^{N+1} - 1)/(b - 1)$, since the bucket widths form a geometric series. Given the exponential nature of this term, it is possible to capture large magnitude rewards with a relatively small value of $N$. For example, with $b = 2$ and $N = 20$, the algorithm fully captures all rewards of magnitude up to 1 million, which is more than sufficient for *Atari*. Taking this logic one step further, we arrive at the following proposition, which is proved in the Appendix:

**Proposition 1** *In the tabular setting, Spectral Q-learning behaves identically to standard Q-learning, provided that the rewards are bound in magnitude to no more than $(b^{N+1} - 1)/(b - 1)$.*

Now that we have presented the full algorithm, it is worth reflecting on why we use a spectral representation for the rewards instead of, say, a binary encoding. Put simply, very small differences in reward should not matter much to a rational agent. For example, it should not matter much whether a particular reward's magnitude is 63 or 64. However, under a binary encoding, 63 is represented as 0111111, while 64 is represented as 1000000. Since a reward of 63 activates more frequencies than a reward of 64, it would thus trigger a larger TD error at the start of training. The thermometer-style encoding introduced earlier addresses this problem.

## 5  SPECTRAL DQN

In order to create a deep learning version of the Spectral Q-learning algorithm, we modify the DQN algorithm in an analogous way to which we modified the Q-learning algorithm in the previous section. We name the resultant algorithm *Spectral DQN*.

To parameterise all $N + 1$ frequencies of the spectral action-value function via a single network, we multiply the number of outputs on the standard DQN network by the same count. One crucial detail here is that we initialise the weights and biases of the final layer to zero. This allows us to set $N$ to a conservatively large value, since it ensures that superfluous heads have no effect on training. Moreover, it ensures that the action-value calculations do not blow out due to the exponential weights in Equation 7 being applied to random noise.

For the training loss, we use a weighted sum of the temporal difference errors for each frequency:

$$\mathcal{L}(\theta) = \tfrac{1}{2} \sum_{i=0}^{N} w_i \left[ r_b^i + \gamma \max_{a'} \tilde{Q}(s', a', i; \theta^-) - \tilde{Q}(s, a, i; \theta) \right]^2 \qquad (8)$$

where $w_i$ is the error weighting given to the $i$th frequency, $s$ is a state sampled uniformly from the replay memory, $a$ is the sampled action, $s'$ is the next state, $r$ is the reward (with spectral components $r_b^i$ for $i \in \{0, \ldots, N\}$), $\theta$ are the parameters of the network being trained, and $\theta^-$ are the parameters of the periodically refreshed target network.

An important detail here is the approach we take to weighting the loss for each frequency. Given that the $ith$ frequency is weighted by $b^i$ when calculating the overall action-values, it is tempting to set $w_i = b^i$. *However, this goes completely against one of the aims of our approach, which is to prevent the large reward frequencies from dominating the loss.* In fact, we show in the next section that weighting by $b^i$ has a disastrous impact in the two extreme domains introduced earlier.

In light of the above, rather than weighting by $b^i$, we strive to give all frequencies equal weighting. However, we do not simply set $w_i = 1$, because doing so would effectively give too much emphasis to the *small* reward frequencies. As pointed out by van Hasselt et al. (2016), the gradients propagated to the Q-network's hidden layers depend quadratically on the scale of the targets (see Section 2.3 of their work). Since the lower frequencies have denser rewards, and hence larger training targets, they will dominate the gradients at the hidden layers if this effect is not accounted for. To address this, we set $w_i = 1/\sigma_i^2$, where $\sigma_i$ is the standard deviation of the training targets for frequency $i$, calculated as a running average as in the Pop-Art algorithm. Finally, to ensure that this correction only affects the hidden layers, we unscale the gradients at the final layer to train as if $w_i = 1$, as in standard DQN. (Implementing this is not especially difficult; the precise details can be found in our source code.) In Appendix B.1, we provide an ablation where we do not correct the hidden layer gradients, and we find that this has a significant adverse effect on performance.

## 6  RETURNING TO THE EXTREME DOMAINS

To test how well Spectral DQN meets our aims, we first applied it to the two extreme domains from Section 3 – *Ponglantis* and *Exponential Pong* – where reward progressivity is clearly a key issue. In addition, to probe the importance of the frequency loss weighting, we included a variant of Spectral DQN with exponential loss weighting, i.e. $w_i = b^i$, which we argued against in the previous section. The results of this experiment, including the baseline results from Section 3, are shown in Figure 3, with each curve averaged over 5 seeds.

There are two clear findings here: (1) Spectral DQN is much better at handling extreme reward progressivity than the baselines. It scores much higher in *Ponglantis*, and makes far more meaningful progress in *Exponential Pong*, eventually learning to beat the computer player comfortably on

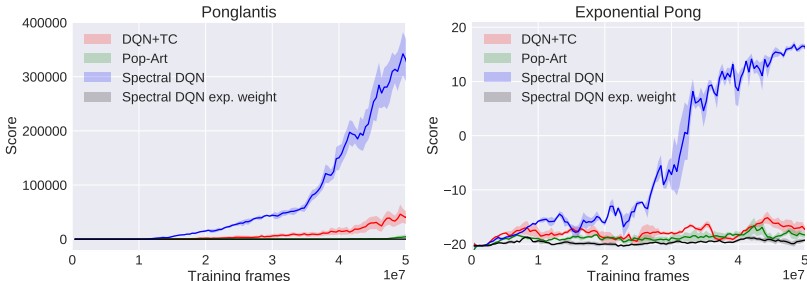

Figure 3: Mean episodic return for Spectral DQN vs. various baselines on the two extreme domains.

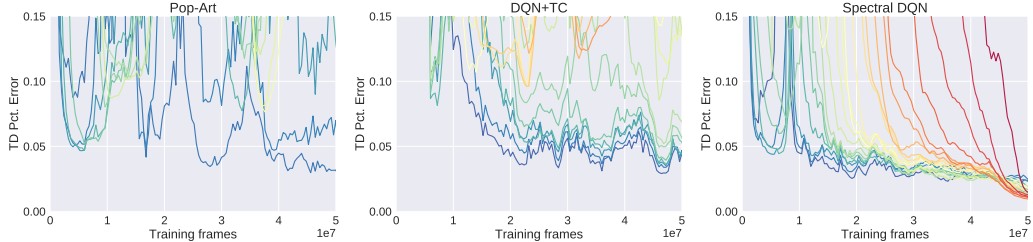

Figure 4: Evolution of the temporal difference percentage error in *Exponential Pong*.

average. (2) The loss weighting is of crucial importance, with the agent using exponential weights performing worse than even DQN+TC. The latter finding matches our expectations; as we noted in Section 3, target compression does at least somewhat mitigate the overemphasis given to large reward regions of the task, while Spectral DQN with $w_i = b^i$ gives heavy emphasis to such regions.

In Figure 4, we examine how the agents' training errors evolve as they play *Exponential Pong*. The errors are broken down by the player's current score $\in [0, 20]$, with bluer lines corresponding to states with a low player score and redder lines corresponding to states with high player scores. To make the graphs comparable, we plot percentage TD errors, i.e.:

$$mean(|Q_{predicted} - Q_{target}|) \, / \, mean(|Q_{target}|) \tag{9}$$

where the means are running averages, calculated from states sampled from the replay buffer. In all cases we report the true TD error, i.e. for DQN+TC we report the error on the uncompressed action-values, and for Spectral DQN we report the error on the full, summed action-values.

The instability of Pop-Art is quite apparent here: After a very brief period of stable learning, the TD errors start spiking as soon as the agent reaches even middling scores. Consistent with our earlier analysis, DQN+TC appears a lot more stable. For states with low player scores, the TD percentage errors hover at around 5%, although the agent never really learns accurate action-values for states where the player score exceeds $\approx$7 points. At first this might seem strange – since the reward scale increases exponentially with the player's score, states with a high player score ought to be effectively prioritised by the loss function. However, there is an additional effect to consider here: As soon the agent's predictions start to deteriorate for the lower-score states, its policy for these states likewise deteriorates and it starts struggling to reach the higher-score states. As such, the agent seems to get stuck in a middle ground, where it does not encounter the higher-score states often enough to learn accurate action-values there, nor for the estimates in the lower-score states to truly deteriorate. It appears that a TD percentage error of around 5% is a crucial cutoff line, where the agent is right on the cusp of being able to play well, but playing better, conversely, leads to deterioration.

In contrast to both Pop-Art and DQN+TC, when Spectral DQN reaches a new high score, it quickly learns accurate action-value estimates for the associated states, as indicated by the sharply dropping red lines in Figure 4. Moreover, the TD errors for the lower-score states remain stable as the agent improves, almost certainly because Spectral DQN's loss does not give so much weight to large magnitude rewards. Finally, note that the agent's sudden rise in performance at around $2.5 \times 10^7$ frames in Figure 3 seems to coincide with the errors reducing below the 5% line, adding further credence to the analysis above.

# 7 EXPERIMENTS IN STANDARD ATARI GAMES

To validate that Spectral DQN's success in extreme domains does not come at the expense of performance in less constructed tasks, we also applied it to 6 standard *Atari* games. Moreover, we included 3 games where reward clipping is known to be detrimental, as per van Hasselt et al. (2016): *Bowling*, *Centipede* and *Ms. Pacman*. This was done to validate that the agent's earlier success was not attributable to it somehow performing an implicit form of reward clipping. The results of these experiments are shown in Figure 5, with all curves averaged over 5 seeds.

While Spectral DQN's performance relative to the baselines is not as exceptional here as in the extreme domains, it remains the strongest overall. Though the agent underperforms DQN+TC in *Centipede*, it noticeably outperforms both baselines in *Bowling*, *Chopper Command* and *Video Pinball*. On the other games, it performs at least on par with the benchmarks. Moreover, the fact that Spectral DQN's performance compares favourably to Pop-Art in *Bowling*, *Centipede* and *Ms. Pacman* shows that it is truly responding to the unclipped reward.

Since *Video Pinball* exhibits reward progressivity, as explained in the introduction, Spectral DQN's strong performance there fits well with our hypothesis. On the other hand, while *Ms. Pacman* also has progressive rewards — the player receives an increasing bonus for eating multiple ghosts with a single power pellet — these rewards come so close together that they are within the agent's temporal horizon at the start of an episode. That is, there is not a small-expected-return phase preceding a large-expected-return phase, which we hypothesised to be a central component of the issue. (We return to this point and discuss further in the next section.)

Besides addressing reward progressivity, another possible advantage of Spectral DQN in high scoring games is simply that its training targets are more constrained. In games such as *Video Pinball* and *Chopper Command*, the expected returns can reach into the thousands, so even with target compression this may be problematic for DQN+TC. On the other hand, the only game where DQN+TC clearly outperforms Spectral DQN is *Centipede*, which is also high scoring. One possible explanation for this is that the reward structure in *Centipede* "confuses" Spectral DQN: Since large rewards trigger both low and high frequencies, it is likely that Spectral DQN learns a positive association between the various components of the return. However, in *Centipede*, losing a life triggers a stream of consecutive small rewards. Therefore, associating small and large rewards may cause Spectral DQN to predict an inflated return from dying.

There has been some conjecture that an advantage of combining distributional RL (Bellemare et al., 2017; Lyle et al., 2019) with function approximation is that it forces the agent to learn a richer representation. The same argument may also apply to our approach. In particular, in some tasks, small rewards are of more informational value than their face value represents. For example, in *Video*

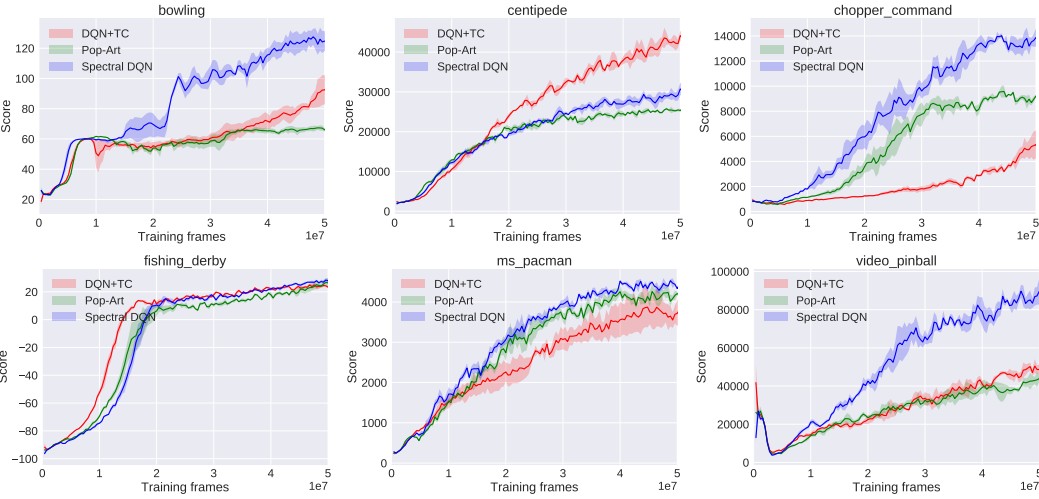

Figure 5: Mean episodic return for Spectral DQN vs. baselines on standard *Atari* games.

*Pinball*, where rewards range into the thousands, the play field includes "spinners" that pay only one point per hit. Despite their negligible face value, hitting the spinners can dramatically change the velocity of the ball, so giving the lower frequencies more weight in the training loss may promote a stronger representation. Similarly, in *Bowling*, certain frequencies only activate after bowling strikes and spares. Since the game does not have a strongly progressive reward structure, improved representation learning is a more plausible explanation of Spectral DQN's outperformance.

## 8    DISCUSSION AND CONCLUSION

In this paper, we brought attention to the idea of reward progressivity, a task property that poses a potential issue for value-based reinforcement learning agents. Specifically, an agent's performance in relatively unrewarding regions of a task may degrade once it discovers larger rewards. To address this issue, we proposed Spectral DQN, which decomposes the expected return magnitudinally, allowing the losses arising across different regions of the task to be balanced. Our experiments showed this approach to be particularly effective in domains exhibiting extreme reward progressivity, while still yielding more than competitive performance in less constructed domains.

Our discussion of the *Ms. Pacman* results from the previous section raised an interesting point: There is clearly a link between reward progressivity and discounting. If it were somehow possible to train reliably with $\gamma = 1$, a *Ponglantis* agent would be able to see that reaching the large, distant rewards in the *Atlantis* phase is contingent on first performing well in *Pong*. However, despite some recent progress, DQN-style agents typically destabilise with discounts very close to 1. *Agent57* (Badia et al., 2020a) trains multiple value functions across a range of discounts, ranging as high as $\gamma = 0.9999$. However, the authors note that $\gamma = 0.9999$ generally yields very unstable learning, and they rely on a meta-controller to select smaller discounts in games where learning destabilises. As such, it does not offer a general solution to the problem. The idea of setting $\gamma = 1$ also raises the issue of reward sparsity: If the agent is cognizant of large, distant rewards, then smaller, more immediate rewards may appear so negligible that the task effectively becomes a sparse reward problem.

We noted in Section 3 that another idea for addressing reward progressivity is to apply target compression, but with logarithmic squashing instead of a square root. However, as Pohlen et al. (2018) note, the inverse of a logarithmic mapping is not Lipschitz continuous, and thus the transformed Bellman is no longer guaranteed to be a contraction. On the other hand, one way to get around poor convergence in practice is to use a shaper discount factor. Van Seijen et al. (2019) show that it is possible to achieve performance comparable to DQN via a logarithmic squash, provided the discount is small ($\gamma = 0.96$). However, they clip the rewards, which greatly constrains the gradient of the inverse mapping $h^{-1}$, and thus limits the degree to which their implementation transgresses the theory. Moreover, the performance of their method degrades significantly with $\gamma = 0.99$, consistent with convergence issues (see Appendix C.2 in their work). Unfortunately, it is simple to provide examples where $\gamma = 0.96$ will lead to poor performance. For example, in the game Bowling, there is a delay of about 100 frames between the ball being released and the subsequent reward. However, if the player bowls a spare, the reward is delayed by a further 100 frames. Thus, under a discount of 0.96, it is better to consistently bowl 9s than to bowl spares.

Our work fits into a broader line of research whereby reinforcement learning agents seek to predict quantities other than just the mean expected return (Jaderberg et al., 2016; Bellemare et al., 2017; Fedus et al., 2019). Of this related work, one of the most similar approaches to ours is that of Romoff et al. (2019), who learn the value function in a modular way by decomposing based on the discount factor. Distributional reinforcement learning (Bellemare et al., 2017; Dabney et al., 2018b;a) is another closely related approach. Finally, an older line of work in this vein is decomposed RL (Russell & Zimdars, 2003; Sprague & Ballard, 2003; Van Seijen et al., 2017), whereby an agent is responsible for predicting only a portion of the action-value function. While some of these works bear many surface-level similarities to our own — the distributional *C51* agent (Bellemare et al., 2017) in particular bears quite a similar network architecture to ours — on a deeper level, they are not addressing the same problem. Leveraging a probabilistic loss, as in C51, could foreseeably help to address the large training errors that arise in tasks like *Exponential Pong*. However, the method assumes that returns are constrained to [-10, 10], and it is unclear how this could be extended to unclipped rewards without requiring an intractable number of atoms in the distribution, or greatly reducing the precision of the atoms.

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

## A  EXPERIMENTAL SETUP

The source code for our experiments is available at `https://github.com/mchldann/SpectralDQN`.

We injected stochasticity into the *Atari* environment via the no-ops regime used in the Nature DQN paper (Mnih et al., 2015). Rather than conducting periodic evaluations of the agents with a small exploration constant, we simply report their training performance, since this arguably provides a more reliable measure of progress. All curves in the paper were averaged over 5 seeds. Shaded regions represent bootstrapped single standard deviation confidence intervals.

Hyperparameters specific to Spectral DQN are listed in Table 2, while hyperparameters common to all agents are listed in Table 3. Spectral DQN used an identical network architecture to Nature DQN Mnih et al. (2015), except that we multiplied the number of outputs by $N + 1$.

| Hyperparameter | Value |
|---|---|
| Base of spectral decomposition, $b$ | 2.0 |
| Maximum frequency, $N$ | 20 |

Table 2: Hyperparameters specific to the Spectral DQN agent.

| Hyperparameter | Value |
|---|---|
| Optimiser | Adam |
| Adam learning rate | $2.5 \times 10^{-5}$ |
| Adam regularisation constant | 0.005 / 32 |
| Discount factor | $0.99^{1/3}$ |
| Multi-step backup $n$ | 3 |
| Final exploration $\epsilon$ | 0.01 |
| Reward clipping | None |
| Life loss signal used | True |

Table 3: Hyperparameters shared between all agents.

## B  ADDITIONAL EXPERIMENTS

In this section, we provide the details of some additional experiments that had to be omitted from the main text due to space constraints.

### B.1  ABLATION: NO ERROR WEIGHTING

Recall from Section 5 that we weight the frequencies' individual losses by $w_i = 1/\sigma_i^2$, where $\sigma_i$ is the standard deviation of the training targets for frequency $i$. In Figure 6, we show the effect of ablating this adjustment and simply setting $w_i = 1$. The results reveal that the adjustment is a crucial component, with the ablation proving disastrous in 4 of the 6 games. (Note: It unsurprising that the *Fishing* results are unaffected, because only one reward frequency is active in that game.)

### B.2  EASIER PONGLANTIS

In Section 3, we applied Pop-Art and DQN+TC to the game *Ponglantis*. The results suggested that DQN+TC was somewhat more robust to reward progressivity. Here, we provide some further experiments to determine just how strong the reward progressivity has to be for these agents to start deteriorating.

To hone in on the agents' breaking points, we created two modified versions of the game: *Ponglantis Easier* and *Ponglantis Even Easier*. In *Ponglantis Easier*, the agent's perceived rewards in the *Atlantis* phase of the task are scaled down by a factor 10. In *Ponglantis Even Easier*, they are scaled down by a factor of 100. The agents' results in these games are shown in Figure 7. For comparability,

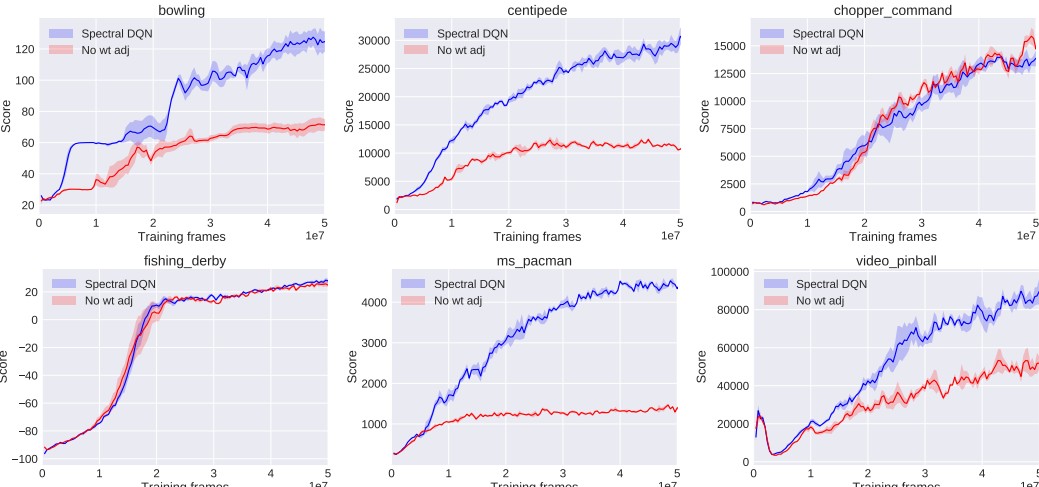

Figure 6: Mean episodic return for Spectral DQN vs. Spectral DQN with no weight adjustment ($w_i = 1$) on standard *Atari* games.

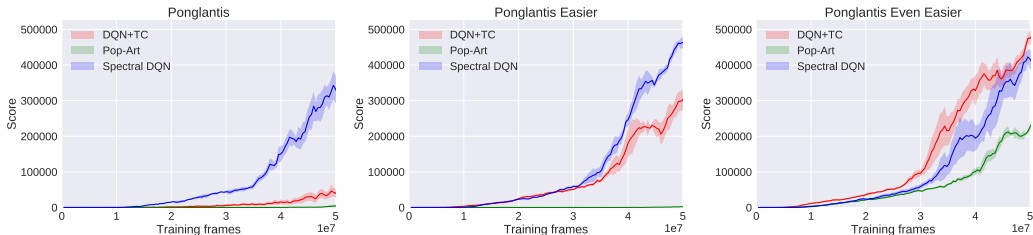

Figure 7: Mean episodic return for agents in *Ponglantis*, *Ponglantis Easier* and *Ponglantis Even Easier*.

the plots show the true episodic return, rather than the perceived episodic return. We also include results for Spectral DQN as a reference point.

Relative to its performance in the original *Ponglantis* task, DQN+TC performs markedly better in both easier tasks. Since the rewards jump by a factor of around 1000 in the original task, this suggests that DQN+TC can tolerate a jump in reward magnitude of around 100x. Further, this implies that DQN+TC's underperformance relative to Spectral DQN in the main Atari experiments is attributable to more than just reward progressivity, as none of the standard Atari games exhibit this level of jump. Our alternate hypotheses about representation learning and the relative ease with which the spectral architecture can represent large expected returns thus appear more likely.

Pop-Art, on the other hand, still performs poorly when the perceived rewards in *Atlantis* are scaled down by a factor of 10, only learning reliably when they are scaled down 100 times. In this case, it is plausible that the algorithm struggles with reward progressivity in standard Atari games.

### B.3 REVERSE PONGLANTIS

To test whether it is truly reward *progressivity* that is the key issue in *Ponglantis*, as opposed to just reward *variability*, we created one further task, *Reverse Ponglantis*, where the *Pong* and *Atlantis* phases are reversed. The player progresses to the *Pong* phase upon losing a life in *Atlantis*.

The agents' results in this game are plotted in Figure 8. While all agents improve steadily, which confirms that progressivity is the main issue for Pop-Art, it is notable DQN+TC's performance is actually worse here than in *Ponglantis Even Easier*. This is despite the fact that the agents now start in the high-scoring *Atlantis* phase, so there is no need to play *Pong* well. From this result, it is clear that a key issue for DQN+TC is simply the reward scale in *Atlantis*. This adds further weight to our

hypothesis from Section 7 that one of the main advantages of Spectral DQN in standard Atari games is the constrained nature of its training targets.

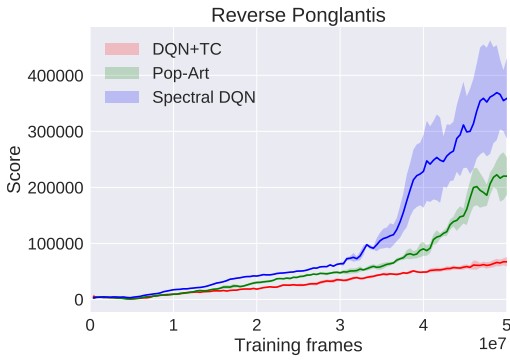

Figure 8: Mean episodic return for agents in *Reverse Ponglantis*.

## C    SPECTRAL Q-LEARNING ALGORITHM

Pseudocode for the Spectral Q-Learning algorithm is provided in Algorithm 1.

---

**Algorithm 1** Spectral Q-Learning

---

1:  **var:** the base of the spectral decomposition, $b$
2:  **var:** the number of spectral components trained, $N$
3:  **var:** the learning rate, $\alpha$
4:
5:  Initialise $Q(s, a, i)$
6:  **for** each episode **do**
7:      Sample $s$ from start distribution
8:      **while** $s$ is not terminal **do**
9:          Choose $a$ according to $\epsilon$-greedy policy derived from $Q(s, a) = \sum_{i=0}^{N} b^i Q(s, a, i)$
10:         Take action $a$, observe $r, s'$
11:         $a'^* = \arg\max_{a'} \sum_{i=0}^{N} b^i Q(s', a', i)$
12:         **for** $i \in \{0, \dots, N\}$ **do**
13:             $Q(s, a, i) \leftarrow Q(s, a, i) + \alpha[r_b^i + \gamma Q(s', a'^*, i) - Q(s, a, i)]$
14:         $s \leftarrow s'$

---

## D    DERIVATION OF THE SPECTRAL DECOMPOSITION FORMULA

Recall from Section 4.1 that the spectral representation of a number is found by laying buckets of exponentially increasing width on the number line, then calculating the proportion of each bucket that is filled.

To calculate the proportion of the $i$th bucket that is filled by a non-negative number $r$, note that the sum of all the preceding buckets' widths is equal to:

$$\sum_{k=0}^{i-1} b^k = \frac{b^i - 1}{b - 1} \tag{10}$$

Therefore, the amount leftover to fill the $i$th bucket is:

$$r - \frac{b^i - 1}{b - 1} \tag{11}$$

To calculate the proportion of the $i$th bucket that is filled, we must divide this quantity by the width of the $i$th bucket (which is equal to $b^i$) then clamp the result to the allowable range of values:

$$\text{Filled proportion} = clamp\big((r - \tfrac{b^i-1}{b-1})/b^i, 0, 1\big) \tag{12}$$

Finally, to convert this into an odd function so that the representation of $-r$ is equal to the negated representation of $r$, we replace $r$ with $|r|$ and multiply the result by $sign(r)$. This yields the formula for the $i$th element of the spectral representation:

$$r_b^i = sign(r) \times clamp\big((|r| - \tfrac{b^i-1}{b-1})/b^i, 0, 1\big) \tag{13}$$

# E  PROOFS

## E.1  PROOF OF EQUATION 7

Starting with the standard definition of the action-value function, then replacing the reward at each time step with its spectral decomposition, we have:

$$Q^\pi(s,a) = \mathbb{E}_\pi[\textstyle\sum_{k=t}^\infty \gamma^{k-t} r_k \mid a_t = a, s_t = s] \tag{14}$$

$$= \mathbb{E}_\pi[\textstyle\sum_{k=t}^\infty \gamma^{k-t} \sum_{i=0}^\infty b^i r_{b,k}^i \mid a_t = a, s_t = s] \tag{15}$$

Since there is no interaction between the summation indices, we can rearrange (15) into:

$$Q^\pi(s,a) = \mathbb{E}_\pi[\textstyle\sum_{i=0}^\infty b^i \sum_{k=t}^\infty \gamma^{k-t} r_{b,k}^i \mid a_t = a, s_t = s] \tag{16}$$

Finally, using the linearity of expectation, we can take the expectation operator inside and write:

$$Q^\pi(s,a) = \textstyle\sum_{i=0}^\infty b^i \mathbb{E}_\pi[\sum_{k=t}^\infty \gamma^{k-t} r_{b,k}^i \mid a_t = a, s_t = s] \tag{17}$$

$$= \textstyle\sum_{i=0}^\infty b^i Q^\pi(s,a,i) \tag{18}$$

## E.2  PROOF OF PROPOSITION 1

**Proposition 1** *In the tabular setting, Spectral Q-learning behaves identically to standard Q-learning, provided that the rewards are bound in magnitude to no more than $(b^{N+1} - 1)/(b-1)$.*

We prove this statement via induction. The first point to observe is that, provided the action-value estimates of the two algorithms agree at a given point in time, their $\epsilon$-greedy policies at that time will also be equivalent. Therefore, assuming that the action-value functions agree at $t = 0$ (e.g. they are both initialised to zero), all we need to prove is the inductive step, i.e. that agreement in action-values at time $t$ implies agreement in action-values at time $t + 1$.

To distinguish between the action-value estimates of the two algorithms at different time steps, we use the following notation:

- Let $\hat{Q}(s,a;t)$ denote the action-value estimates under standard Q-learning after $t$ updates.

- Let $Q(s,a;t) = \sum_{i=0}^N b^i Q(s,a,i;t)$ denote the action-values estimates under Spectral Q-learning after $t$ updates.

For the induction step, we need to show that $Q(s,a;t+1) = \hat{Q}(s,a;t+1) \ \forall s \in \mathcal{S}, a \in \mathcal{A}$ given that $Q(s,a;t) = \hat{Q}(s,a;t) \ \forall s \in \mathcal{S}, a \in \mathcal{A}$.

To begin, note that the standard Q-learning backup expressed under the above notation is:

$$\hat{Q}(s,a;t+1) \leftarrow \hat{Q}(s,a;t) + \alpha\big[r + \gamma\hat{Q}(s',a'^*;t) - \hat{Q}(s,a;t)\big] \tag{19}$$

where $a'^* = \arg\max_{a'} \hat{Q}(s',a'^*;t)$ is the greedy action for the next state.

Since we assume equal action-values at time $t$, Spectral Q-learning will calculate the same greedy action for $s'$. Hence, the backup for each frequency of the spectral action-value function is:

$$Q(s, a, i; t+1) \leftarrow Q(s, a, i; t) + \alpha \left[ r_b^i + \gamma Q(s', a'^*, i; t) - Q(s, a, i; t) \right] \tag{20}$$

After performing this update on each frequency, the function at time $t+1$ becomes:

$$Q(s, a; t+1) = \sum_{i=0}^{N} b^i \left[ Q(s, a, i; t) + \alpha \left[ r_b^i + \gamma Q(s', a'^*, i; t) - Q(s, a, i; t) \right] \right] \tag{21}$$

$$= \sum_{i=0}^{N} b^i Q(s, a, i; t) + \sum_{i=0}^{N} \alpha b^i \left[ r_b^i + \gamma Q(s', a'^*, i; t) - Q(s, a, i; t) \right] \tag{22}$$

$$= Q(s, a; t) + \sum_{i=0}^{N} \alpha b^i \left[ r_b^i + \gamma Q(s', a'^*, i; t) - Q(s, a, i; t) \right] \tag{23}$$

Since the induction base case assumes that $\hat{Q}(s, a; t) = Q(s, a; t) \ \forall s \in \mathcal{S}, a \in \mathcal{A}$, we can rewrite this as:

$$Q(s, a, i; t+1)$$

$$= \hat{Q}(s, a; t) + \alpha \sum_{i=0}^{N} b^i \left[ r_b^i + \gamma Q(s', a'^*, i; t) - Q(s, a, i; t) \right] \tag{24}$$

$$= \hat{Q}(s, a; t) + \alpha \left[ \sum_{i=0}^{N} b^i r_b^i + \gamma \sum_{i=0}^{N} b^i Q(s', a'^*, i; t) - \sum_{i=0}^{N} b^i Q(s, a, i; t) \right] \tag{25}$$

$$= \hat{Q}(s, a; t) + \alpha \left[ \sum_{i=0}^{N} b^i r_b^i + \gamma Q(s', a'^*; t) - Q(s, a; t) \right] \tag{26}$$

$$= \hat{Q}(s, a; t) + \alpha \left[ \sum_{i=0}^{N} b^i r_b^i + \gamma \hat{Q}(s', a'^*; t) - \hat{Q}(s, a; t) \right] \tag{27}$$

Now, all that remains to establish equality with the standard Q-learning update in (19) is to show that $\sum_{i=0}^{N} b^i r_b^i = r$. That is, we need to show that the partial sum over indices 0 to $N$ fully captures the reward, or, equivalently, that $r_b^i = 0$ for all $i >= N+1$.

From the spectral decomposition formula (13) and the proposition's statement that the rewards are bound to no more than $(b^{N+1} - 1)/(b - 1)$, we have:

$$|r_b^i| = clamp\left( (|r| - \tfrac{b^i - 1}{b - 1})/b^i, 0, 1 \right) \tag{28}$$

$$\leq clamp\left( (\tfrac{b^{N+1} - 1}{b - 1} - \tfrac{b^i - 1}{b - 1})/b^i, 0, 1 \right) \tag{29}$$

$$= clamp\left( (\tfrac{b^i(b^{N+1-i} - 1)}{b - 1})/b^i, 0, 1 \right) \tag{30}$$

Now, if $i >= N+1$, it implies that $(b^{N+1-i} - 1) \leq 0$, so the whole expression in (30) becomes 0.

Thus $\sum_{i=0}^{N} b^i r_b^i = \sum_{i=0}^{\infty} b^i r_b^i = r$ and the proof is complete. ∎

