# OpenReview forum: "Adapting to Reward Progressivity via Spectral Reinforcement Learning"
_ICLR.cc/2021/Conference — ICLR 2021 Poster_

### Official Review · AnonReviewer3 · 2020-10-21
**Simple, effective, and intuitive -- but lacking analysis**

**Rating:** 7
**Confidence:** 4

**Review:**

This work describes and addresses the issue of _reward progressivity_ in reinforcement learning, where as the task progresses the scale of the reward changes. The authors argue that reward progressivity harms Q-learning when training signals arising from large rewards interfere with those arising from smaller rewards. They propose a form of reward decomposition with an analogous modification to the Q-network output, which together help to ensure that training losses from small and large rewards are similarly scaled. The authors present a handful of experimental results demonstrating that their proposed method outperforms two other reward re-scaling baselines when reward progressivity is an issue and maintains good performance in more standard tasks.


### Clarity
This paper is exceptionally clear. There are a couple areas where techincal details are somewhat missing: namely, their implementation of the mixed Monte Carlo update. Otherwise, the ideas are very clearly presented and the authors are careful to spell out caveats and details, which will improve reproducibility and downstream adaptation.

### Originality and Quality
_Spectral DQN_ is, to my knowledge, original. However, comparison to other forms of reward decomposition is lacking, both experimentally and in the discussion. The authors mention several examples of works that enable non-scalar representation of rewards for handling variability but there is not much attention given to how  these prior works are different and/or why they are not included as baselines.

_Reward progressivity_ as a motivating issue is also, to my knowledge, original. Indeed, the authors point out the 2 main baselines do not consider _intra-task_ reward variability. This brings up one of my main issues with the paper: there is very little demonstration that reward progressivity is the disruptive force that the authors describe it as. The authors show that baselines suffer/fail on tasks that are engineered towards this problem, but this doesn't offer enough to convince a sceptical reader to believe the paper's underlying intuitions. My understanding of these intuitions is that reward progressivity causes RL to stagnate because it biases the Q-network to be accurate around large rewards at the expense of being accurate around low rewards -- the issue being that this latter inaccuracy leads to poor policies in important but low-reward regions of the task.

If this is indeed the intuition, then it would be very useful to actually demonstrate some sort of accuracy trade-off induced by reward progressivity. This would also provide a useful lens through which to validate Spectral DQN, by perhaps being able to characterize how it balances Q-network accuracy for different reward scales compared to other baselines.

It would also be valuable to discuss whether reward progressivity is ultimately an issue of discounting. The authors seem to touch on this when describing the _Mrs. Pacman_ results, but a bit more exposition would be useful.

There are numerous other examples where the authors add interpretation around a particular result to help characterize where Spectral DQN does/doesn't succeed and what a practitioner might want to look out for.These interpretations improve the paper's quality somewhat, but the level of analysis is ultimately lacking. For instance, the authors offer speculation over the results for _Centipede_ and _Bowling_ but don't back up this speculation with any meaningful analysis. I would not want to punish the authors for offering these insights (that commentary is valuable and honest) but I do think the paper would have been better served by spending more time on analysis to back up a couple key insights.

### Significance
If we take the authors' claims at face value (which I am still inclined to do even though the paper lacks more detailed validation), this paper presents a simple technique to handle reward progressivity. This contribution is significant from a practical standpoint but perhaps only when solving tasks with this particular reward structure. For this to have broader signficance, it would be necessary for the authors to provide a more detailed characterization of the potential pitfalls associated with the technique's heuristics. Again, these are discussed (which is appreciated) but only speculatively.


**Pros**
- Adds a simple technique for balancing losses associated with rewards of different
 scales.
- Follows a straightforward intuition.
- Overcomes extreme examples of the problem it is designed to address.
- No obvious, systematic drawbacks of the proposed modification.
- Exceptionally clear and well-written.

**Cons**
- Analysis is limited to a handful of training curves with only one minor ablation (in Figure 3)
- The paper tends to emphasize speculation over more concrete demonstration.
- Unclear whether the motivating problem presents a significant obstacle.

### Suggestions for improvement
I would be happy to increase my score if the authors are able to address my criticisms. In particular, I think an analysis demonstrating the how learning dynamics are negatively impacted by reward progressivity (via its measurable effects on the Q-network) would go a long way. In addition, a more thorough ablation study and/or sensitivity analysis would add very useful clarity around the impact of, for example, the mixed Monte Carlo update strategy.


### Post-discussion feedback
Thank you for the valuable discussion and revisions. I believe the paper has improved and I've updated my score to reflect that.

The new analysis in Figure 4 and its surrounding discussion offer some useful insights. I think the authors could still improve this analysis a bit in a final version of the paper (if only just to make the plots a bit easier to parse visually), but the TD-error instability of the baselines seems reasonably clear and I'm inclined to agree with how the authors frame this as an example of the problem they solve. In addition, the added discussion around discounting is also valuable; and the simplification of the algorithm and added ablation experiments improve the quality of the contributions.

---

> ### Author Response · Authors · 2020-11-20
> **Author response**
>
> Thank you for your considered review. Since some of your concerns were also raised by other reviewers, please see our general comments to all reviewers before reading the below.
>
> **“The authors show that baselines suffer/fail on tasks that are engineered towards this problem, but this doesn't offer enough to convince a sceptical reader to believe the paper's underlying intuitions.”**
>
> Thank you for explaining this criticism in such detail, it is a fair point. We are currently running some additional experiments of the type you asked for, plus some extra ablations, and we will have more to say about these once the results are in.
>
> **“It would also be valuable to discuss whether reward progressivity is ultimately an issue of discounting.”**
>
> We only touched on this in the current version of the paper due to space constraints, but we will make use of the extra page allowed for the next version to discuss this in more detail.
>
> In our view, yes, reward progressivity is ultimately an issue of discounting. If it were somehow possible to train without a discount, i.e. gamma = 1, then the expected returns (and hence the temporal difference errors) ought to be largest at the *beginning* of Exponential Pong and Ponglantis. Put another way, the problem with discounting in progressive reward tasks is that it can cause “indifference”; if the large rewards late in the task are discounted too heavily, the agent will not care sufficiently about performing well in the early stages of the task in order to reach them.
>
> Unfortunately, increasing gamma towards 1 is not a viable solution, because algorithms like DQN start to destabilise with very mild discounts. In our experiments, we already pushed gamma to 0.99^(1/3), which triples the agent’s temporal horizon compared to vanilla DQN. (As noted in the Appendix, we use n-step learning with n = 3 to maintain the contractiveness of the backup operator.) However, in our experience, increasing the discount beyond this point starts to cause instability.
>
> **“comparison to other forms of reward decomposition is lacking, both experimentally and in the discussion.”**
>
> We have responded to this in part (C) of our general comments, since Reviewer 2 raised a related point about distributional RL.

---

> > ### Comment · Area_Chair1 · 2020-11-23
> > **Further questions on the relevance of discounting**
> >
> > The reviewer question and the author response indicate that the interaction between discounting and reward structure is important for this problem. The authors also state that it was difficult to train a DQN-like system with larger discounts. Some previous papers have demonstrated techniques for training DQN-like systems with larger discounts, such as the Agent-57 paper (https://arxiv.org/pdf/2003.13350.pdf) and "Separating value functions across time-scales" (https://arxiv.org/pdf/1902.01883.pdf).
> >
> > Can the authors expand on why increasing gamma to one is not viable? Does the problem identified in this paper also arise when gamma is one? The author response says the expected returns should be largest at the start of an episode, but this is not true when there are both negative and positive rewards. Is the proposed method intended primarily for deep RL architectures that have been developed for small discount factors?

---

> > > ### Author Response · Authors · 2020-11-23
> > > **Increasing gamma to one can work in select tasks, but doesn't generalise.**
> > >
> > > Hi, thanks for the clarifying questions.
> > >
> > > In ongoing tasks, it is not possible to set gamma = 1, because the returns may become unbounded. While the returns are bounded in episodic tasks, with gamma = 1 the Bellman operator is no longer a contraction, so it may take a long time for action-values with inaccurately large magnitudes to compress. The only reliable "source of truth" to contract the estimates is episode termination. This can be problematic under deep function approximation, as is perhaps best illustrated via an example:
> > >
> > > Suppose that the task is a simple video game, where an enemy ship enters from the top of the screen that the player must shoot. Upon shooting the ship, the player receives 100 points, and a new ship enters the screen. Suppose that the screen looks more or less the same each time the new ship enters, so the sequences of states looks something like s_0, s_1, s_2, ... s_0. In other words, there is a loop from s_0 back to itself, with a positive reward in between. If gamma = 1, then the net effect of learning from this loop will be to push the action-values up. Now suppose that there is a very small, constant chance of the ship's gun backfiring and killing the player, such that the task is now episodic. There will now be a single, very rare TD error that pushes the action-values down. However, if we use gradient clipping, or a Huber loss, or some optimizer that prevents huge parameter updates via an adaptive denominator term (e.g. Adam), then the impact of this rare TD error may not be enough to offset the inflation in the other direction. In other words, it's not so much a theoretical issue as a practical one, which is probably why there isn't a definitive reference on this kind of problem, at least as far as we know of. For gamma < 1, the problem is mitigated, since once the action-values reach a great enough magnitude, the discounting will offset the positive rewards. (Side note: The loop from s_0 to s_0 need not be *exact*; the start and end states just have to be close enough that the agent estimates very similar action-values.)
> > >
> > > Regarding the papers you mentioned, the paper "Separating value functions across time-scales" uses an overall discount of 0.99 in their Atari experiments. (This is actually less than the discount of 0.99^(1/3) that we use.) They only use larger discounts in their tabular experiments, where the practical issues mentioned above don't arise. Regarding Agent-57, what we should have said is that deep RL agents don't learn *consistently* with discounts very close to 1. In particular, note their comment on page 8: "running R2D2 with a high discount factor, gamma = 0.9999, surpasses the human baseline in the game of Skiing. However, using that hyperparameter across the full set of games, renders the algorithm very unstable and damages its end performance". If a task's rewards are very sparse (e.g. a long task with a single reward of 1 upon successful completion), then practical issues like the above are less likely to arise. However, this is not a general solution, and Agent-57 relies on a meta-controller to select smaller discounts on the (majority) of games where gamma = 0.9999 is unstable. Thus the issue of reward progressivity is still relevant on the whole.
> > >
> > > Regarding both positive and negative rewards, you are correct. (This is why we gave the examples of Exponential Pong and Ponglantis, where optimal policies only receive positive rewards. We were being a bit inexact to simplify the argument, though we should have been clearer.) All we really mean is that under gamma = 1, the actions at the start of the episode are much more *consequential*, since they affect all future rewards. Thus the issue of "indifference" that we described in our previous response ought to be mitigated.

---

### Official Review · AnonReviewer1 · 2020-10-22
**Review for Spectral RL**

**Rating:** 7
**Confidence:** 3

**Review:**

## Summary

This paper details the problems that might arise in value-based reinforcement learning methods in domains where reward progressivity is present. To show that current methods do not handle reward progressivity, the authors introduce two domains, _Exponential Pong_ and _Ponglantis_.

After showing that current methods do not work well in these domains, the paper then goes on to propose a solution, spectral decomposition of rewards. The paper shows that returns learned separately on decomposed rewards can be composed to get the original return. The paper then presents spectral Q-learning and spectral DQN, with experiments on the domains presented earlier as well as experiments on 6 Atari games.

On the two domains where progressive rewards were shown to be problematic, Spectral DQN is shown to work better than current approaches. On Atari games, spectral does as well as current approaches, doing better in 3 out of 6 domains.

## Positives
+ The setting of progressive rewards is interesting.
+ The domains proposed for testing these rewards are clear.
+ The spectral reward decomposition is a simple idea and is explained well.
+ The effectiveness of spectral DQN on the two domains introduced in the paper is clear.
+ Experimental details are clear and additional steps taken to stabilize learning are included.

## Negatives/ Questions
- One question that does not seem to be satisfactorily addressed is whether there is a commonly used benchmark domain or one which was not specifically engineered for progressive rewards where value based deep reinforcement learning would fail or not perform well unless it was using Spectral DQN.
- The spectral DQN objective (eqn. 8) includes target compression. It is unclear how much benefit the spectral decomposition is having in the presence of target compression. While it is fine to require target compression for the full benefit of spectral DQN, it would be good to see an ablation of how well spectral DQN does without target compression.
- Another useful ablation could be to remove the monte carlo mixing and show how unstable the updates get.
- How does the van Seijen et al. paper on logarithmic mappings (Using a Logarithmic Mapping to Enable Lower Discount Factors in Reinforcement Learning) compare to the related work? It is motivated by a different problem, but since a possible logarithmic mapping might mitigate the problems that come up with progressive rewards their method might be a possible solution to be compared against.

## Other Comments
* Figure 2 has a typo (Fiilled instead of filled).

## Summary
Overall, I find the idea in this paper clear, simple, and effective. There are some additional questions and comments that if addressed would make the paper a more well-rounded submission.

---

> ### Author Response · Authors · 2020-11-20
> **Author response**
>
> Thank you for your considered review. Since some of your concerns were also raised by other reviewers, please see our general comments to all reviewers before reading the below.
>
> **Is there a “commonly used benchmark domain or one which was not specifically engineered for progressive rewards where value based deep reinforcement learning would fail or not perform well unless it was using Spectral DQN”**
>
> Since Reviewer 2 raised a similar point, we have addressed it in part (A) of our general comments.
>
> **“It is unclear how much benefit the spectral decomposition is having” / “Another useful ablation could be to remove the monte carlo mixing and show how unstable the updates get.”**
>
> We appreciate these points, and we are running additional experiments to address them. We will have more to say soon when the results are in.
>
> **How does the van Seijen et al. paper on logarithmic mappings (Using a Logarithmic Mapping to Enable Lower Discount Factors in Reinforcement Learning) compare to the related work?**
>
> Thank you for making us aware of this work, it is indeed relevant and we will update the paper to cite it. Like us, it seems that the authors were strongly influenced by Pohlen et al. (2018). Interestingly though, their approach of using a logarithmic mapping is contradictory to Pohlen et al.’s requirement that the mapping function have a Lipschitz continuous inverse in order to ensure convergence. (See our discussion at the end of page 3, where we explain why we didn’t use a logarithmic mapping.) Practically, one way to get around poor convergence is to use a shaper discount factor, and this is indeed what van Seijen et al. do. Note that they use a discount of 0.96 in their main experiments, but that the performance of their method degrades significantly with a discount of 0.99, consistent with convergence issues (see Appendix C.2 in their work). The paper frames the enablement of a sharp discount as an advantage, but in reality a sharp discount comes with a big drawback; namely, it greatly constrains the agent’s temporal horizon. It is easy to provide examples where this will lead to poor performance. For example, in the game Bowling, there is a delay of about 100 frames between the ball being released and the subsequent reward. However, if the player bowls a spare, the reward is delayed by a further 100 frames. Thus, under a discount of 0.96, it is better to consistently bowl 9s than to bowl spares. The reason why their Bowling results remain on par with DQN is because both methods use reward clipping, and all clipped reward methods perform terribly in this game. (See their Figure 14, and note that a score of 30 implies that the agent is bowling 3s on average.)
>
> Given its violation of the Lipschitz constraint, we were initially surprised to find a convergence proof in van Seijen et al.’s work. However, the proof should be taken with a grain of salt, since one of its requirements is for the step size to decay to zero. (Specifically beta_2 must go to zero, but beta_1 * beta_2 is analogous to the standard step size -- see the bottom of page 8 in their paper.) It is much easier to ensure convergence under this condition, but it is a significant practical limitation and van Seijen et al. do not actually decay the step size to zero in their experiments.
>
> **Figure 2 has a typo (Fiilled instead of filled).**
>
> Thanks for picking this up!

---

### Official Review · AnonReviewer2 · 2020-10-23
**An intriguing idea, well-developed, somewhat lacking a use case**

**Rating:** 6
**Confidence:** 4

**Review:**

This paper proposes an extension to DQN, more generally applicable to value-based deep RL systems, that encodes the return using a thermometer encoding with exponentially-sized bins. This enables returns of vastly differing magnitudes to be learned without hurting performance. The authors propose an algorithm for learning these encode returns, including the use of a variance scaling term to speed up learning.

Overall, I enjoyed reading the paper and appreciated the clear exposition, which is sensible throughout. My main concern is whether this is solving a real problem, or a hypothetical one. The experiments don't support the former case, while I would argue that solving the problem hypothetically would require a more thorough development. For example, if progressivity is an issue, what form do we expect it to take? The two synthetic examples, Ponglantis and Exponential Pong, make for a fun case but not necessarily a realistic one. Training on all Atari games with a single network, for example, might be a better case.

I have a few technical questions for the authors:

4.1: Why use a thermometer encoding versus a binary encoding? I.e., write the reward in binary, and encode its bits.

4.2: Why not use base gamma instead of base 2? That seems like a more natural way to encode returns.

4.3: "impossible to train over full, infinite spectrum" -- given that returns are bounded for gamma < 1, what do you mean? It seems like you need log(VMax) bins at most.

5: "We do not simply set ..." I would have expected some ablation studies here, regarding the role of the parameter. The choice w_i = 1/sigma_i seems particularly ad-hoc. In particular, I'm disappointed that you did not include the experiment w_i = 1. It seems natural to me that exponential weights should fail, whereas it's not clear that w_i = 1 should, and the use of sigma complicates things (and adds a moving part).

The results of Fig. 4 are not terribly exciting. Sometimes spectral DQN works better, sometimes it does not. Can you comment on this? Should we be concerned? Also, you should include published DQN results (and if possible, other value-based methods) as I would expect these to be worse for games that have rewards of different magnitudes.

8: "the phenomenon of reward progressivity". Is the issue really progressivity? Or is it that value-based learners struggle to deal with rewards of varying magnitude (the progression doesn't matter)?

Distributional RL algorithms replace the L2 loss with probabilistic losses. For example, the C51 algorithm uses the KL divergence. That could help deal with reward scale also, by creating a loss that is insensitive to scale. In particular, I can imagine the binary cross-entropy loss making sense since most of your outputs are binary.


Minor points

- Condition on gamma < 1/L_h L_h-1 -- doesn't that mean that Lh must be almost 1? Not much slack.
- Intro, 2nd paragraph. A concrete example would help the reader here.
- Table 1 might want to be left justified, as it's a little jarring to read.


==== Updated review

In light of the authors' revisions, I'm happy to raise my score to 6. I think the paper is better, although I still wish the presentation was more compelling -- as given, this seems more like an exercise than a contribution with a demonstrated impact. On the other hand, this level of contribution seems relatively on par with e.g. other deep RL papers.

Regarding the revisions, I would encourage the authors to integrate them with the main text. For example, Figure 3 really wants the weight=1 result (maybe as two separate panels -- comparison to other algorithms, a); ablation on weights, b)).

---

> ### Author Response · Authors · 2020-11-20
> **Author response (1/2)**
>
> Thank you for your considered review. Since some of your concerns were also raised by other reviewers, please see our general comments to all reviewers before reading the below.
>
> **“My main concern is whether this is solving a real problem, or a hypothetical one.” Also: “The results of Fig. 4 are not terribly exciting. Sometimes spectral DQN works better, sometimes it does not. Can you comment on this?”**
>
> Since Reviewer 1 also raised this point, we have addressed it in part (A) of our general comments.
>
> **“Why use a thermometer encoding versus a binary encoding? I.e., write the reward in binary, and encode its bits.”**
>
> We actually touch on this in the Appendix (see the last paragraph of page 11), but since it is an important point we will make it more prominent. Essentially, it would be weird for the agent to treat rewards of very similar magnitude in a completely different way. For example, it should not matter much to a rational agent if a particular reward’s magnitude is 63 or 64. However, under a binary encoding, 63 is 0111111, while 64 is 1000000. A reward of 63 would thus trigger a much larger TD error at the start of training than a reward of 64 (since 63 triggers many more frequencies). The thermometer encoding addresses this problem.
>
> **“Why not use base gamma instead of base 2? That seems like a more natural way to encode returns.”**
>
> Base gamma might be a natural way to encode returns if our central problem was estimating how far away the rewards are, in terms of time steps. However, to deal with very large rewards, our approach requires a base greater than one, to ensure that the buckets have increasing width. (Without this, it would not be possible to represent rewards of arbitrary size with the thermometer encoding.) Another option would be to use base 1/gamma, but for bases very close to 1 we would require a huge number of frequencies to fully capture all rewards in Exponential Pong.
>
> **“given that returns are bounded for gamma < 1, what do you mean? It seems like you need log(VMax) bins at most.”**
>
> Our only point here is that you don’t always know VMax in advance. That being said, we show that this isn’t a big limitation, since with exponentially sized buckets it’s possible to represent very large rewards with reasonably small N. Side note: We’re not sure whether it is clear, but the number of heads required is actually determined by the maximum *reward* magnitude, not the maximum return. Please let us know if we can clarify this.
>
> **“The choice w_i = 1/sigma_i seems particularly ad-hoc. In particular, I'm disappointed that you did not include the experiment w_i = 1.”**
>
> This is a good point. We actually did run some initial experiments with w_i = 1, but found that the agent’s performance was adversely affected. We will elaborate on the reasons for this in the revised paper. Moreover, we agree that this should be established properly via experiments, and we are now running a full ablation.
>
> **“you should include published DQN results (and if possible, other value-based methods) as I would expect these to be worse for games that have rewards of different magnitudes.”**
>
> Interestingly, as discussed in the Pop-Art paper, this is not always the case. It turns out that the reward clipping mechanism is a very good heuristic for Atari, and disabling it sometimes leads to worse performance. (There is a nice discussion about why this happens on page 8 of the Pop-Art paper. In particular, see their discussion of the game Time Pilot.) Exponential Pong also provides a good example -- if we were to apply standard DQN to this game then it ought to do very well, because standard DQN sees no difference between Exponential Pong and standard Pong. However, the way it achieves this (reward clipping) is unprincipled, which is why we don’t consider DQN as a benchmark in this paper.
>
> **"the phenomenon of reward progressivity". Is the issue really progressivity? Or is it that value-based learners struggle to deal with rewards of varying magnitude (the progression doesn't matter)?**
>
> A good illustration of this point is Ponglantis. If the Altantis phase happened to come before the Pong phase, then Pop-Art and DQN+TC would both get decent scores. (Both methods perform well on standard Atlantis, so they will do well on the first phase of the task. It doesn’t matter if they subsequently fail on Pong, because the bulk of the points come from the Atlantis phase.) From this example, it’s clear that the problem only comes to the fore if the large reward phase comes *after* the small reward phase, i.e. the rewards are progressive. We will include an experiment with reverse Ponglantis to properly establish this.

---

> > ### Author Response · Authors · 2020-11-20
> > **Author response (2/2)**
> >
> > **“Distributional RL algorithms replace the L2 loss with probabilistic losses. For example, the C51 algorithm uses the KL divergence. That could help deal with reward scale also, by creating a loss that is insensitive to scale.”**
> >
> > Since Reviewer 3 raised a similar point, please see part (C) of our general comments.
> >
> > **“Condition on gamma < 1/L_h L_h-1 -- doesn't that mean that Lh must be almost 1?”**
> >
> > It’s possible that the compression function could just be a linear rescaling, in which case L_h = 1 / L_h-1. So you could have, for example, Lh = 5, L_h-1 = 0.2. However, you are correct that for non-trivial scaling functions, this is quite a strict requirement. It may be possible to derive a more relaxed bound, but we are not aware of any work to this end.
> >
> > **Other minor points**
> >
> > Thank you for bringing these to our attention, we will address them in the next draft.

---

### Official Review · AnonReviewer4 · 2020-10-28
**Review of 'Adapting to reward progressivity via spectral RL'**

**Rating:** 6
**Confidence:** 3

**Review:**

#######################################################################

Summary:

In this paper the authors propose a new RL method, spectral DQN, in which rewards are decomposed into different frequencies. This decomposition allow for the training loss to better balanced on certain tasks - in particular those with progressive rewards. The new method is shown to perform well on specially constructed tasks with extreme reward progressively, as well as on a selection of standard Atari tasks.

#######################################################################

Reasons for score:

I think a weak accept is appropriate here. I think the authors correctly identify a class of worth while tasks - namely those with progressive rewards, and reasonably establish that standard approaches struggle here. The new method is a strong implementation of a simple idea which is demonstrated to work, and does not require significant fine-tuning.

#######################################################################Pros:

1. The paper is well written and clearly presented. I'd commend the authors on their exposition and balanced motivation throughout

2. I think this is an interesting direction of work more generally - exploring the performance of different methods against different reward distributions, and having agents predict quantities more flexible than the mean return. This would seem to be a contribution to that body of work.

3. I see no reason that the constructed domain ExponentialPong shouldn't join the benchmark for any subsequent general-purpose agent 🙂

#######################################################################

Cons:

1. While I believe the selected experiments are sufficient to demonstrate the claims in the paper, they do not fully explore the capabilities of the method and the intuitions that motivate it. It would have been good to see some more thinking here (even if it means experiments outside of Atari)

(1) It would seem that this approach would work for a parametrizable class of reward functions - why not test that? Perhaps ExponentialPong with reward $b^(\alpha N)$ evaluated on a grid for $b$ and $\alpha$?

(2) Similarly for failure modes

(3) Frequencies needn't be geometric, how might other choices have performed

2. I'd like to have seen a slightly richer discussion/motivation of the mixed Monte Carlo update. Clearly something is necessary here, but why this? What else was tried?

#######################################################################

Questions during rebuttal period:

Q1: you note the obvious limitation in the selection of the number of reward frequencies, but as you note this is reasonably easily ameliorated in practice. A more interesting question perhaps is whether an adaptive approach for b might work - was this tried?

#######################################################################

Some typos:

(1) 'Expoential Pong' → 'Exponential Pong', just above 4.1'

(2) Fig2, last row, "Fiilled proportion" → 'Filed proportion'

---

> ### Author Response · Authors · 2020-11-20
> **Author response**
>
> Thank you for your considered review. Since some of your concerns were also raised by other reviewers, please see our general comments to all reviewers before reading the below.
>
> **“I see no reason that the constructed domain ExponentialPong shouldn't join the benchmark for any subsequent general-purpose agent”**
>
> Thank you for your encouragement, we agree!
>
> **“While I believe the selected experiments are sufficient to demonstrate the claims in the paper, they do not fully explore the capabilities of the method and the intuitions that motivate it.”**
>
> This is a fair criticism, and we are currently running a number of additional experiments in order to provide a better exploration of the method. We will have more to say about this once the results are in.
>
> **“It would seem that this approach would work for a parametrizable class of reward functions - why not test that? Perhaps ExponentialPong with reward b^(alpha x N) evaluated on a grid for b and alpha?”**
>
> Evaluating on a grid for b and alpha would require a lot of compute, but we agree that it would be helpful to include some additional experiments to find where the “cutoff point” is, i.e. how big the jump in reward magnitude has to be for Pop-Art and DQN+TC to start breaking down. To address this, we are running some additional experiments in Ponglantis, where we have scaled down the reward in the second, Atlantis phase of the task.
>
> **“Frequencies needn't be geometric, how might other choices have performed.**
>
> The issue here is tractability. With quadratically scaling frequencies (b_i = i^2), the approach would require almost 150 frequencies to capture rewards up to 1 million, as in Exponential Pong. If we could somehow know in advance that the rewards do not scale exponentially, then a sub-exponential scale would be appropriate. However, to make Spectral DQN as general as possible, we didn’t want to rely on this kind of knowledge.
>
> **“I'd like to have seen a slightly richer discussion/motivation of the mixed Monte Carlo update. Clearly something is necessary here, but why this? What else was tried?”**
>
> We too felt that the mixed Monte Carlo update was slightly unsatisfactory, and have since devised a cleaner method where we just initialise the weights and biases of the final layer to zero. We are running fresh experiments now, and will provide more commentary once the results are in.
>
> **Was an adaptive approach for b tried?**
>
> This is indeed an interesting idea, as it would allow us to cut down on “wasted” frequencies in tasks where the rewards scale sub-exponentially. However, this would require the neural network’s outputs to be adjusted when b is changed, so as to preserve the old predictions (similar to the way the Pop-Art algorithm works). Unlike Pop-Art, something much more sophisticated than a linear scale and shift would be required though, which is an interesting problem for future work.
>
> **Typos**
>
> Thank you for picking these up, we will address them in the next draft of the paper.

---

### Author Response · Authors · 2020-11-20
**General comments (1/2)**

We thank the reviewers for the time and effort they have invested in reviewing our paper. Since there was some overlap in the questions and concerns raised by the reviewers, we thought it would help to address those in a general post. Other queries will be addressed below the individual reviews.

**(A) Is Spectral DQN solving a real problem, or a hypothetical one?**

Spectral DQN is addressing a real problem. However, a reason why reward progressivity might appear to be more of a hypothetical issue is because RL research has, to date, focussed mostly on single task learning. Since rewards are often relatively steady intra-task, reward progressivity is often not such a big problem. However, if we ultimately wish to train agents in multi-task settings then reward progressivity will need to be addressed. Progressive rewards are not the only difficulty here; for example, to train a single DQN-style agent on all Atari games, the replay memory would need to be expanded substantially, and the runtime would increase beyond our computational means. While don't have an answer to all of these issues in this work, we believe that reward progressivity is still an important piece of the puzzle.

While there have been some large-scale attempts to train a single network to play all games in the Atari-57 suite, e.g. IMPALA (https://arxiv.org/abs/1802.01561), to the best of our knowledge they all clip rewards to [-1, 1]. As the Ponglantis experiments show, unclipping the reward is disastrous for Pop-Art and DQN+TC when even two of the games from this suite are combined.

We agree with Reviewer 2 that our standard Atari experiments, in and of themselves, do not strongly establish the “realness” of the problem. However, to clarify, that was not exactly what these experiments were intended to show. We do not expect the issue of reward progressivity to arise in all domains, but it does in some. As such, we thought it was important to establish that Spectral DQN is at least non-detrimental in less handpicked tasks, and the standard Atari experiments do support this. Moreover, while the results were less clear-cut than in the extreme domains, Spectral DQN did notably perform well in games with mild reward progressivity (Chopper Command, Ms Pacman and Video Pinball).

**(B) Too many moving parts, lack of ablations.**

In Section 5, we mention three measures that we used to stabilise Spectral DQN:

(1) Weighting the ith frequency’s error by w_i.
(2) Using target compression on the individual frequencies.
(3) Adaptive Mixed Monte Carlo updates.

The overall feeling we got from the reviews is that this constitutes too many moving parts, and that extra ablations would help. In short, we agree, and we have been working on simplifying the approach. While our results are not ready to share yet, we have devised a cleaner approach that does away with measures (2) and (3). Moreover, we are running a full ablation to show the importance of (1). We will update the paper and provide more commentary on this when the results are in, but for now we just wanted to provide the reviewers with this update.

---

> ### Author Response · Authors · 2020-11-20
> **General comments (2/2)**
>
> **(C) “The authors mention several examples of works that enable non-scalar representation of rewards for handling variability but there is not much attention given to how these prior works are different and/or why they are not included as baselines”**
>
> This is an important point, so we will expand on it in the next revision. The reason we have cited these works is because they bear many surface-level similarities to our method. However, on a deeper level, they are not actually addressing the same problem. Some of them are not related to reward variability at all; for example, Romoff et al.’s (2019) method for decomposing the expected return with respect to the time scale is unrelated to reward variability.
>
> Reviewer 2 notes that the probabilistic loss used by the distributional C51 algorithm might help address reward variability. However, it is important to note that the C51 algorithm learns from the clipped reward, and it assumes that the expected return lies in [-10, 10]. While reward clipping is one way to deal with reward variability, it is unprincipled. To handle unclipped rewards and deal with games where the expected return can reach tens of thousands, one would either need to increase the number of return buckets substantially, so as to cover a wider range, or increase the size of the buckets. The first of these options is computationally infeasible. The second option would result in a loss of precision, and it was shown in the C51 paper that the performance of the algorithm degrades significantly when the bucket size is increased by even a factor of 10 (see Figure 3 in the C51 paper).
>
> The other main type of distributional algorithm -- namely, implicit quantile methods -- still ultimately learns expected returns rather than probabilities. As such, we cannot see any reason why it would be especially well-suited to progressive reward tasks. The reason we benchmarked against DQN+TC, on the other hand, was because its square root compression *could* foreseeably help with the problem.
>
> Van Seijen et al. (2017) propose an approach where a human expert provides a decomposed reward function and the agent learns different value functions for the different rewards. This could potentially help with the problem (e.g. in Ponglantis, we could train one value function for Pong, and one for Atlantis), but the need for a human expert is a big drawback.

---

### Author Response · Authors · 2020-11-24
**Revision Summary**

Hi all,

Just letting you know that we've uploaded a revised version of the paper with new results. We're still making a few edits, but we wanted to leave a small window in case anyone has questions about the new experiments.

Summary of the main changes:
- We've included some additional analysis of what is happening behind-the-scenes as the agents train in Exponential Pong (page 7).
- We've included an ablation of the error weighting adjustment across all six Atari games, showing what happens when w_i = 1. (See Appendix B.1. TLDR: The weight adjustment is a crucial component.)
- We've included some additional experiments in variants of Ponglantis to find where "cutoff point" for Pop-Art and DQN+TC is, i.e. how big the jump in reward magnitude has to be for these agents to start breaking down. (See Appendix B.2.) We've also included a "Reverse Ponglantis" experiment, per our feedback to Reviewer 2. (See Appendix B.3.)
- We've greatly simplified the implementation, removing both the adaptive mixed Monte Carlo update and the target compression on individual frequencies. The agent now performs noticeably better in Chopper Command and Video Pinball, and comes much closer to learning an optimal policy for Exponential Pong. (The key to this simplification was realising post submission that the weight adjustment ought to be w_i = 1/sigma_i^2, rather than w_i = 1/sigma_i. This is not an ad hoc choice; it has a proper theoretical underpinning, as now explained on page 6. Target compression was previously masking the issue, because it meant that the various sigma_i's were closer in value. After making this change, we found that the mixed Monte Carlo updates and target compression were no longer necessary.)
- We have expanded Section 8 to provide a more in-depth discussion of the relationship to discounting. We also explain why various recent methods still do not offer a general, principled approach to dealing with reward progressivity.

Thanks again for your all feedback and discussion, we feel that it has significantly improved the paper.

---

### Author Response · Authors · 2021-03-12
**Source code**

Source code for our agent is available at:
https://github.com/mchldann/SpectralDQN

---

### Decision · Program_Chairs · 2021-01-07
**Final Decision**

**Decision:**

Accept (Poster)

**Comment:**

This paper presents a deep RL algorithm to handle tasks where rewards can differ greatly in magnitude.  The proposed solution decomposes the reward into a set of exponentially sized bins with a thermometer encoding, and computes a weighted sum of the value functions learned for each bin.  The approach addresses the common tactic of reward clipping and value rescaling in deep RL algorithms.  The experiments demonstrate the potential utility of this approach on artificially constructed Atari games, and the experiments also show the approach remains competitive on six standard Atari games.

The reviewers found both strengths and weaknesses in the paper.  The overall approach was viewed as a clear and sensible (R1, R2, R4) approach to handling widely varying reward scales in a domain.  It may be a useful contribution in a manner similar to other methods that make deep RL algorithms more robust to scaling issues encountered in practice (R4).  The main concerns were whether this was solving a real problem or not (R2, R3), and the lack of a theoretical development for the multiple heuristics (R2,R3).

The author response then simplified the algorithm, which also served to clarify which aspects of the algorithm were relevant to the performance improvements. The response removed some of the heuristics (mixing in Monte Carlo returns) and changed other choices to be more principled ($1/\sigma^2$). The author response also described how the proposed algorithm addressed different scaling concerns from those handled by earlier methods. The author response also provided clarifications to many minor questions raised by the reviewers.  In the ensuing discussion, the reviewers were happy with the revised paper.  Though some minor theoretical reservations remained, the reviewers agreed this paper was a useful contribution.

The reviewers indicate to accept the paper as a useful contribution in deep RL to address certain reward scaling issues.  The paper is therefore accepted.